# Methodologies for Remote Bridge Inspection—Review

**DOI:** 10.3390/s25185708

**Published:** 2025-09-12

**Authors:** Diogo Ribeiro, Anna M. Rakoczy, Rafael Cabral, Vedhus Hoskere, Yasutaka Narazaki, Ricardo Santos, Gledson Tondo, Luis Gonzalez, José Campos Matos, Marcos Massao Futai, Yanlin Guo, Adriana Trias, Joaquim Tinoco, Vanja Samec, Tran Quang Minh, Fernando Moreu, Cosmin Popescu, Ali Mirzazade, Tomás Jorge, Jorge Magalhães, Franziska Schmidt, João Ventura, João Fonseca

**Affiliations:** 1iBuilt, School of Engineering, Polytechnic of Porto, 4249-015 Porto, Portugal; drr@isep.ipp.pt (D.R.); rps@isep.ipp.pt (R.S.); jorbm@isep.ipp.pt (J.M.); jpsva@isep.ipp.pt (J.V.); jpdrf@isep.ipp.pt (J.F.); 2CONSTRUCT-iRail, Faculty of Engineering, University of Porto, 4200-465 Porto, Portugal; 3Department of Civil Engineering, Warsaw University of Technology, 00-637 Warsaw, Poland; anna.rakoczy@pw.edu.pl; 4Department of Civil and Environmental Engineering, University of Houston, Houston, TX 77004, USA; vhoskere@central.uh.edu; 5ZJU-UIUC Institute, Zhejiang University, 718 East Haizhou Road, Haining 314400, China; narazaki@intl.zju.edu.cn; 6Department of Modelling and Simulation of Structures, Faculty of Civil and Environmental Engineering, Bauhaus-Universität Weimar, 99423 Weimar, Germany; gledson.rodrigo.tondo@uni-weimar.de; 7ISISE, Department of Civil Engineering, University of Minho, 4800-058 Guimarães, Portugal; luismgonzalez@uvigo.gal (L.G.); jmatos@civil.uminho.pt (J.C.M.); jtinoco@civil.uminho.pt (J.T.); minhtq@civil.uminho.pt (T.Q.M.); 8Department of Structural Engineering and Geotechnics, University of São Paulo, São Paulo 05508-220, Brazil; futai@usp.br; 9Department of Civil and Environmental Engineering, Colorado State University, Fort Collins, CO 80521, USA; yanlin.guo@colostate.edu; 10Civil and Environmental Engineering Department, Rowan University, Glassboro, NJ 08028, USA; trias@rowan.edu; 11Bridge and BIM Consultant, Austria; vsamec7@gmail.com; 12Department of Civil, Construction & Environmental Engineering, University of New Mexico, Albuquerque, NM 87106, USA; fmoreu@unm.edu; 13Sintef Narvik AS, 8517 Narvik, Norway; cosmin.popescu@sintef.no; 14Invator AB, 181 22 Lidingö, Sweden; ali.mirzazade@invator.se; 15MAST-EMGCU, Université Gustave Eiffel, Champs-sur-Marne, 77420 Paris, France; franziska.schmidt@ifsttar.fr

**Keywords:** methodologies, remote bridge inspection, computer vision, Big Data, Digital Twins, Augmented Reality

## Abstract

This article addresses the state of the art of methodologies for bridge inspection with potential for inclusion in Bridge Management Systems (BMS) and within the scope of the IABSE Task Group 5.9 on Remote Inspection of Bridges. The document covers computer vision approaches, including 3D geometric reconstitution (photogrammetry, LiDAR, and hybrid fusion strategies), damage and component identification (based on heuristics and Artificial Intelligence), and non-contact measurement of key structural parameters (displacements, strains, and modal parameters). Additionally, it addresses techniques for handling the large volumes of data generated by bridge inspections (Big Data), the use of Digital Twins for asset maintenance, and dedicated applications of Augmented Reality based on immersive environments for bridge inspection. These methodologies will contribute to safe, automated, and intelligent assessment and maintenance of bridges, enhancing resilience and lifespan of transportation infrastructure under changing climate.

## 1. Introduction

With the aging of infrastructure assets worldwide, it has become crucial for infrastructure managers to adopt efficient and reliable methods of evaluation of their structural condition. This evaluation helps to extend the expected serviceability of infrastructure and to ensure adequate levels of safety. In this review, Structural Health Monitoring (SHM) refers to the sensor-based systems that enable continuous or periodic measurements of structural behavior. In contrast, remote bridge inspection involves the visual assessment of surface conditions using technologies such as different types of camera sensors mounted on UAVs (Unmanned Aerial Vehicles) and LiDAR (Light Detection and Ranging). While monitoring captures structural responses that are used to detect early-stage damage (visible and non-visible) and their evolution over time, inspection focuses on detecting visible damage in discrete time periods. These complementary approaches contribute to early damage detection, structural condition assessment, and informed maintenance interventions.

The digitalization of inspection procedures aids analysis and decision-making in bridge management. The digitalization process typically employs methodologies such as computer vision (CV), Big Data (BD), Digital Twins (DT), Augmented and Virtual Reality, and drive-by strategies. These methodologies, with their significant relevance, are combined to realize a complex digitization process for bridges.

Recent advances in computer vision automate key aspects of structural inspection by transforming real-world data into actionable digital insights, thereby reducing the need for human intervention [1]. Current research has focused on developing techniques that support automation, including (i) 3D geometric reconstitution of bridges for Bridge Information Models (BrIM) [2,3]; (ii) automatic damage identification supported by AI [4,5,6]: (iii) non-contact measurement of structural parameters such as displacements [7,8], modal parameters [9,10], and strains [11,12].

Given the volume and complexity of the generated data, BD strategies are essential to enable these applications. These strategies include collection, processing, and analysis of large and complex datasets to produce precise real-time models of bridges, constantly updated with new data into interacting DT [13]. This provides a dynamic tool for bridge maintenance, where predictive analyses based on BD can be employed to identify and address structural issues before they become critical.

DTs typically include several modules: (i) data acquisition and control of the physical assets; (ii) simulation, optimization, and prediction analyses of the virtual model; and (iii) a visualization interface. The simulation module is typically based on numerical/analytical models on computing clouds, reproducing scenarios relevant to decision-making. Optimization algorithms calibrate virtual models. Predictive analysis estimates the future condition of an asset based on its previous conditions to tailor condition-based maintenance. The recent works on DT aim to explore novel DT frameworks to analyze and assess the need for maintenance and repair interventions.

Inspection and monitoring data can be visualized in immersive systems using Virtual Reality (VR) and Augmented Reality (AR) platforms to conduct virtual visits to assets. VR provides a computer-simulated environment, while AR overlays content onto real-world views. Within the Metaverse environment, inspectors can examine inaccessible areas and interact with other inspectors. They can also use AR devices, including smart glasses, to visualize structural data on-site in real time, making it easier to identify and assess potential issues [14]. VR allows engineers and inspectors to immerse themselves in a fully digital 3D model of a bridge. This can be used for remote inspections, training, and simulating various scenarios, such as stress tests or environmental impacts, without the need for physical access to the bridge. Linked-data approaches can also merge a photogrammetric model of the asset and monitoring data. Recently, studies have shown the capability of AR to visualize the mode shapes of structures [15,16]. These technologies enhance decision-making, reduce inspection time, and improve the overall accuracy of bridge assessments by providing more detailed and interactive visualizations of the structure. However, the use of such computer graphics environments for human-centric condition assessment is still limited. Table 1 summarizes the scope of this review, and defines the objective and the techniques used for each methodology.

**Contribution and novelty.** This review focuses on the presentation and comparison of methodologies for remote bridge inspection, in contrast to most existing works in the literature, which concentrate mainly on technologies and platforms [17,18].

## 2. Vision-Based Methodologies

Vision-based approaches offer significant advantages in detecting and evaluating surface defects on bridges, such as concrete cracks, spalling, and steel bar corrosion, among others [19,20,21]. They are also valuable for identifying structural components to provide appropriate contexts for damage assessment [22]. These methods enable early detection of critical conditions, facilitate timely maintenance decisions, and reduce the need for manual work during inspection activities.

This section provides a theoretical framework for computer vision methodologies. Section 2.1 describes 3D bridge reconstruction based on photogrammetry, LiDAR, and hybrid approaches. The section also provides insights into point cloud processing techniques enabling the scan-to-BrIM process and digital BrIM. Section 2.2 details heuristic and deep learning techniques that automatically identify damage from collected images, including integrated frameworks. Section 2.3 describes methodologies for evaluating physical quantities such as displacements, modal parameters, and strains in bridges’ structural elements. Within each subsection, comparative performance assessments of the relevant methods are provided, highlighting trade-offs in accuracy, coverage, and cost.

Figure 1 illustrates the three levels of computer vision methodologies explored in this section.

### 2.1. 3D Geometric Reconstitution

Building Information Modeling (BIM) and BrIM rely on accurate 3D geometric reconstructions. Three-dimensional geometric reconstitution can be performed by photogrammetry, LiDAR, or hybrid approaches which merge the two techniques. These techniques produce point cloud data that can be used to extract information, such as geometry, dimensions of structural components, material types, etc. This information is the basis for generating digital objects constituting BrIM.

#### 2.1.1. Photogrammetry

Photogrammetry is a widely accepted method for the 3D modelling of well-textured surfaces, capable of accurately extracting the 3D shape of an object through strategies such as Structure from Motion (SfM) and Multi-View Stereo (MVS). SfM reconstructs a sparse 3D point cloud and estimates camera positions from a set of unordered images by identifying and matching key features across different views, followed by triangulating those matches to compute the 3D locations of points. Then, MVS uses the estimated camera poses to create a dense point cloud by performing stereo matching on overlapping images.

Photogrammetry algorithms can be used for bridge inspection by reconstructing 3D models of bridges for subsequent segmentation and BrIM generation. This framework has four main phases: (i) site selection and preparation; (ii) image acquisition; (iii) photogrammetry processing to reconstruct a point cloud; and (iv) BrIM generation. Point cloud processing involves data registration, noise removal, segmentation, and feature extraction [23]. Dabous et al. [2] captured 204 high-resolution images, together with calibration data and 56 Ground Control Points (GCPs), to produce a BrIM of a reinforced concrete bridge using Photo Modeler software. They generated four merged point cloud datasets to capture surface detail, optimizing geometric accuracy, model size, and photorealism. Figure 2 presents the BrIM generation process, where Figure 2a shows the merged 3D point cloud, Figure 2b shows the 3D surface model, and Figure 2c shows the corresponding Revit [24] model.

Pepe et al. [3] produced a 3D model of a Roman bridge using photogrammetric algorithms based on UAV and terrestrial images. Point clouds derived from both sources were merged to develop a robust model of the bridge. This study aimed to develop a procedure to obtain a structural analysis bridge model using the Finite Element Method (FEM).

#### 2.1.2. LiDAR

LiDAR is a remote sensing technology that uses laser pulses to measure distances to a target. By emitting pulses of light and calculating the time taken for each pulse to return after reflecting off objects, LiDAR generates 3D point cloud models. This principle is referred to as Time-of-Flight (ToF). Typical LiDAR systems include a 3D laser scanner, Global Navigation Satellite System (GNSS), Inertial Measurement Unit (IMU), and cameras. There are three main types of LiDAR: Terrestrial Laser Scanning (TLS), Mobile Laser Scanning (MLS), and Aerial Laser Scanning (ALS). TLS uses stationary scans; MLS and ALS collect data while in motion.

Hong et al. [25] proposed an automatic method to extract point cloud subsets of surfaces of structural components of a bridge consisting of (i) point-to-surface conversion, (ii) superstructure extraction, and (iii) substructure extraction. The method combines spatial point cloud data with contextual knowledge to sequentially extract subsets corresponding to individual bridge components, progressing from the superstructure down to the substructure. For each bridge component, the two levels of extraction are (1) coarse extraction to separate candidate points of the component from the full dataset and (2) fine filtering via cell- or voxel-based region growing (CRG/VRG), followed by a connected surface component (CSC) step. The authors demonstrated the proposed method on a concrete box-girder bridge in Germany (using five Leica ScanStation P20 scan positions) and a slab beam bridge in the UK (using 18 FARO 3D X 330 scan positions). The method produces a 3D bridge model suitable for management, assessment, and Digital Twin applications. While conventional LiDAR setups are commonly used in structural applications, alternative strategies for 3D imaging have been investigated, such as compressed sensing with dual-frequency LiDAR systems [26].

Tzortzinis et al. [27] used LiDAR to assess section loss on a decommissioned steel I-girder bridge. They scanned a corroded beam to generate a point cloud, which was then used to map the remaining web thickness and analyze its progression over time.

#### 2.1.3. Hybrid Strategies

While photogrammetry offers a low-cost and scalable option for large-area coverage using consumer UAVs, LiDAR-based approaches provide superior accuracy for complex geometries, but they require costly hardware and technical expertise. Combining both methods through point cloud data fusion addresses their individual limitations, enabling a more precise and comprehensive output. Cabral et al. [28] performed a precise hybrid 3D geometric reconstruction of a railway viaduct using both TLS-LiDAR and UAV photogrammetry to facilitate structural inspection and to enable the assessment of geometry and surface damage. TLS gathered data from the lower and lateral deck surfaces (Figure 3a), while a UAV-mounted camera was used to characterize the lateral and upper deck surfaces and the railway track (Figure 3b). The Iterative Closest Point (ICP) algorithm aligns the photogrammetry point cloud with the LiDAR point cloud by estimating a rigid transformation—comprising rotation and translation—that maximizes their spatial overlap. The study resulted in a precise and realistic 3D reconstruction of the entire bridge deck and track, as illustrated in Figure 3c.

Cabral et al. [29] proposed a methodology for combining UAV photogrammetry, TLS, and MLS, within an optimized framework to create a realistic 3D model of a large-scale operational railway bridge. The methodology includes a data cleaning strategy that uses machine learning (CNN SegNet) for background removal through image segmentation in photogrammetry, and a heuristic algorithm for LiDAR data. This approach significantly reduces computational effort by effectively eliminating noise and background data, making it a valuable tool for bridge management. The integrated method effectively addresses the limitations of single-technology surveys, such as incomplete data collection and the lack of texture in laser scanner surveys. The fusion of both datasets results in a detailed and accurate representation of the entire railway bridge (Figure 4).

#### 2.1.4. AI-Based Methodologies

Recent research advances employ Neural Radiance Fields (NeRFs) [30] for 3D reconstruction of civil infrastructure. NeRF learns volumetric representation from input images by training a neural network to predict the color and density at any 3D coordinate and viewing direction. The key idea is that the network can implicitly represent the entire scene by learning how light interacts with surfaces, allowing the synthesis of novel views (i.e., views not captured in the original image set). Bridges often have complex geometries (e.g., arches, cables, and supports), which traditional techniques like SfM or MVS may struggle to capture in detail. NeRF can produce fine geometric features more accurately by modelling the bridge as a volumetric scene, and also by reconstructing surface textures and material properties for detailed visual inspection (e.g., detecting cracks, rust, or corrosion). While NeRF has been predominantly applied to building structures [31], its use in bridge-related contexts has only recently emerged, with initial efforts applied to reduce-scale bridge models. Kim and Cha [32] proposed an integrated framework for Structural Health Monitoring, combining a modified Nerfacto model, an optimized version of NeRF, for the 3D reconstruction and damage mapping of a reduce-scale three-span concrete bridge.

Gaussian Splatting (GS), a cutting-edge neural rendering technique introduced by Kerbl et al. [33], enables real-time novel view synthesis from multi-view images or 3D point clouds. Unlike NeRF models, which represent volumetric scenes, GS renders scenes by projecting collections of 3D Gaussian functions directly onto the image plane. Compared to NeRF models, GS offers significantly faster rendering and improved visual fidelity. Most of its applications have been demonstrated in computer graphics and general scene reconstructions. GS shows promising potential for structural engineering domains such as UAV-based inspections, photogrammetry, and scan-to-BrIM workflows. However, only a few studies have explored the application of GS specifically in the context of bridge inspection.

Another recent approach for characterizing the geometry of structural components with missing or incomplete data is the Plane Fitting Method (PFM), which fits planes to cross-sectional faces of the structural elements. PFM typically employs Random Sample Consensus (RANSAC) algorithm [34]. RANSAC is often combined with the K-means clustering algorithm that uses the normal vector of each point to identify to which face of the section the point belongs. RANSAC requires input parameters, particularly (i) the maximum distance from the points to the model; (ii) the maximum deviation between points’ normal vectors and those of the model, and (iii) the minimum number of points. In a study by Blanco [35], PFM estimated girder cross-section dimensions like web depth and bottom flange width with 0.3% to 1.1% error, while the RANSAC yielded errors of 0.4% to 1.2% error for the same dimensions. Hong et al. [25] proposed an improved method for identifying bridge components by extracting point clouds from sub-datasets and classifying the surfaces as parts of the superstructure and substructure. The authors found that the RANSAC algorithm tends to miss points around edges and can cause over-segmentation, which is suppressed by the proposed method. Figure 5a shows point cloud data for a pier, Figure 5b the result of the RANSAC application, and Figure 5c the segmentation method proposed by the authors.

Deep learning techniques have been applied to the semantic segmentation of point cloud data. Jing et al. [36] developed a neural network, BridgeNet, based on an encoder/decoder framework for segmenting point clouds of bridge components such as piers, girders, and arches. This process is illustrated in Figure 6. The original point cloud (Figure 6a) includes background noise marked for removal. The processed point cloud (Figure 6b) displays a cleaned structure. The segmented point cloud (Figure 6c) distinguishes bridge components by color. In this method, background noise is first removed to discard non-essential components, followed by semantic segmentation. A similar study was presented by Kim et al. [37], where a PointNet neural network was used to segment piers, decks, and background. Bahreini and Hammad [38] used a Dynamic Graph CNN algorithm for surface defect detection using LiDAR sensors. To support such deep learning-based methodologies, Shi et al. [39] introduced a method to create large-scale synthetic point cloud datasets by simulating bridge inspections through a software platform called the Random Bridge Generator [1].

### 2.2. Damage and Component Identification

Damage identification using images is typically based on heuristic techniques (Section 2.2.1) or deep learning (Section 2.2.2). Image-based damage detection can also be incorporated into digital models through dedicated integration frameworks (Section 2.2.3).

#### 2.2.1. Heuristic Techniques

Heuristic feature extraction methods aim to extract meaningful information or features from images without relying on learning-based algorithms. They typically involve (i) image pre-processing, (ii) image registration, and (iii) image classification based on pattern recognition. Image pre-processing involves applying heuristic operators to reduce noise and prepare the images for further stages. Common operators include histogram equalization, sharpening, and local contrast enhancement. Image registration matches two or more images of the same scene, which may be captured at different times (multi-temporal), from different orientations, or even by different types of sensors (multi-modal). This procedure consists of feature detection, feature matching, transformation, and image resampling [40,41]. Finally, pattern recognition groups images with shared features like texture, color, and shape. The approach includes data sensing, segmentation, feature extraction, and classification [42].

The data sensing stage involves image acquisition using various camera sensors (e.g., infrared, RGB, and multispectral). Segmentation applies pattern detection methods. Heuristic feature extraction techniques, such as edge detection and morphological filters, are used to enhance patterns of interest [43]. Edge detection filters like Sobel and Canny can identify component edges [43,44,45]. However, roughness, stains, distinct lighting conditions, distortions, shadows, or blur may result in false positives and reduce segmentation accuracy [46]. To address this limitation, morphological filters help minimize noise and improve edge detection. Common types include dilation, erosion, closing, and opening. These filters can be applied to grow or shrink image regions, as well as to remove or fill in image region boundary pixels [43,47,48]. Potenza et al. [49] proposed a method for damage detection on steel railway bridges using heuristic filters on UAV images. The method was able to efficiently detect and quantify superficial anomalies on structural elements (in terms of area), namely, the lack of paint and the presence of efflorescence and vegetation. The method was validated against expert assessments performed using Computer-Aided Design (CAD) tools and led to significant reductions in manual effort and associated costs. Figure 7 depicts a filter detection of paint loss, including background noise.

Zollini et al. [50] introduced a non-destructive approach for assessing concrete bridge surface degradation. A 3D image-based model was constructed using both terrestrial and aerial images, which was then used to generate orthoimages of the bridge sides. These orthoimages had a fixed pixel size suitable for metric extraction. Each orthoimage was processed using heuristic filters and a Support Vector Machine (SVM) classifier to identify damaged regions. This approach enabled robust and efficient detection and characterization of damage severity, including crack width, length, and total deteriorated area.

#### 2.2.2. Deep Learning

Deep Learning (DL) techniques have proven highly effective in identifying various types of damage in images of structures, including cracks, corrosion, and other structural defects or anomalies. Compared to heuristic methods, which are simple to implement and require minimal training, DL approaches offer significantly higher accuracy and adaptability to diverse damage types. However, these benefits come at the cost of requiring large annotated datasets, computational resources, and expertise in model development and training. Convolutional Neural Networks (CNNs) are the most widely used DL architecture for image analysis due to their ability to automatically learn features from raw pixel data. For damage detection, CNNs are employed to classify, localize, and segment damaged areas. Standard CNNs consist of convolution, activation function (such as Rectified Linear Unit (ReLU) [51]), pooling, and fully connected layers, and can be trained on labelled datasets to detect damaged versus undamaged regions. Pretrained CNNs, like AlexNet, ResNet, or VGG, can be fine-tuned via transfer learning, reducing the need for large, labelled datasets and enabling the identification of specific damage types.

Object detection networks are used to classify and localize damage through bounding boxes. You Only Look Once (YOLO) is a real-time object detection network capable of detecting multiple damage types in a single pass while estimating confidence scores. It is suitable for real-time deployment. Alternatively, Faster R-CNN is a two-stage method that first proposes Regions Of Interest (ROIs) and then classifies the type of damage, providing higher accuracy when precise localization is required. Single Shot MultiBox Detector (SSD) offers faster inference than Faster R-CNN, making it suitable for mobile or edge devices, like drones inspecting infrastructures.

Semantic segmentation techniques assign each pixel by damage type, allowing detailed localization. CNN variants used for this purpose include U-net deep Fully Convolutional Networks (FCN), deep Fusion Convolutional Neural Networks (FCNN), Mask R-CNN, and Deeply Supervised Nets (DSN). Some authors incorporate these segmented results into 3D image models based on dedicated algorithms. Deng et al. [52] used a binocular system with an ADU-Net for bridge crack detection. Fukuoka and Fujiu employed SegFormer, a semantic segmentation framework combining Transformers and Lightweight Multilayer Perception (MLP) decoders. Dung and Anh [53] developed an FCN for crack segmentation, while Alexander et al. [54] used RGB and thermal images to enhance segmentation task. Building upon these advances, Hu et al. [55] combined semantic segmentation using a MobileNetV2_DeepLabV3 network with LiDAR–camera data fusion to enable real-time crack detection and precise 3D localization. This integrated approach allowed for accurate capture of the geometry and spatial distribution of cracks on complex surfaces, achieving sub-millimeter measurement accuracy. Narazaki et al. [22] performed semantic segmentation of bridge components to provide contexts for performing structural assessment. This work was extended in a later study [21], where a large-scale synthetic dataset was used to train high-performance segmentation networks (Figure 8a) and different structural damages (Figure 8b). Kim and Cho [56] introduced one of the first applications of Mask R-CNN to reinforced concrete inspection, demonstrating the advantages of instance segmentation and shifting research focus toward improving detection and segmentation algorithms in complex real-world scenarios.

Additionally, Narazaki applied these algorithms to UAV path planning for autonomous data acquisition [57]. In similar studies, Kim et al. [37,58] focused on bridge component recognition using 3D point cloud data, demonstrating its effectiveness for structural condition assessment.

Among object detection algorithms, YOLO has gained significant attention in the scientific community. Zhang et al. [59] proposed a method for concrete bridge inspection, training their model on 2206 field inspection images covering damage types such as cracks, spalling, pop-outs, and exposed rebars. The authors employed a multiscale training strategy with seven different image sizes to enhance robustness across varying input dimensions. Yu et al. [4] developed a modified Faster R-CNN with a ResNet backbone to detect multiple forms of bridge damage. The authors used 1000 high-resolution images (8688 × 5792 px), some of which contained numerous damage types, which were downscaled to 1086 × 724 px for computational efficiency. K-means clustering was applied to generate anchor boxes for training. Kao et al. [5] utilized YOLOv4 to localize cracks using bounding boxes only. Meanwhile, Sun et al. [60] introduced an optimized YOLOv5 model featuring a decoupled prediction head, CBAM attention module, and focal loss function, achieving improved mAP50 accuracy over earlier models trained on similar datasets [4,59]. More recent YOLO versions (6, 7, 8, and NAS) provided enhanced capabilities for detecting small and overlapping damage, subpixel segmentation, and efficient deployment on mobile or edge devices.

Most automated damage detection tools are based on RGB images, which do not work well in low-light conditions. Hoskere et al. [19] developed MaDnet, a multi-task network capable of segmenting materials (concrete, steel, and asphalt), and detecting both fine and coarse damage (cracks, rebar, and corrosion). MaDnet splits tasks to accommodate different up-sampling filters, making it more efficient than using separate networks, and it can be deployed on UAVs.

In contrast, thermal images maintain consistent performance without degradation but lack the sharpness of detail. A ResNET-based semantic segmentation model was used to fuse RGB and thermal images for vision-based inspections in a study developed by Alexander et al. [54]. A Convolutional Neural Network was employed to identify damage defects in concrete using a thermal and RGB encoder to improve feature detection from both spectra. The results indicated that the RGB-thermal fusion network outperformed the RGB-only network in detecting cracks using the Intersection Over Union (IOU) performance metric. Furthermore, the RGB-thermal fusion model showed better performance in differentiating types of damage and was able to detect damage at a higher rate compared to the RGB-only network.

Steel connections, such as bolts and rivets, play a crucial role in ensuring the structural integrity and long-term safety of steel railway bridges. Lee et al. [61] applied the Mask R-CNN model, with ResNet-101 as the backbone, to detect bolts and signs of corrosion on steel bridge components. The authors originally trained the algorithm using an indoor dataset of 80 images and later fine-tuned it using 100 UAV images from a real bridge survey inspection. The network achieved high detection accuracy, with mAP50 and mAP75 scores of 90% in controlled indoor settings and 89% and 36% under real-world conditions.

DL techniques demand substantial computational resources, and preparing datasets is often time-consuming and labor-intensive. A common solution is to use photo-realistic synthetic environments. As illustrated in Figure 9, this involves (1) defining the geometry with a mesh, (2) preparing texture images, (3) applying those textures, and (4) rendering annotated images using synthetic cameras. These datasets can be used for DL-based visual recognition algorithms, such as structural component/damage recognition [62], and steel crack detection [20].

Narazaki et al. [63] presented an Unsupervised Domain Adaptation (UDA) approach to combine synthetic annotated data and real unlabeled data for crack detection, reducing or eliminating manual labeling time consumption. Synthetic point clouds can also be generated directly through virtual LiDAR sensors, as shown in Figure 10. Rahman et al. [64] used such data for DL-based instance segmentation of bridge components and successfully demonstrated the use of Transformer-based methods on real point clouds.

Despite the progress of DL algorithms in crack detection, the application of Artificial Intelligence remains limited due to the lack of knowledge on crack formation mechanisms. Perry et al. [65] proposed an automated workflow for assessing fracture mechanisms directly from images. A Gaussian Process surrogate model was trained using FEM simulation data to predict stress intensity factors (K), which serve as an indicator of fracture behavior. Using crack dimensions (e.g., length and angle) of identified cracks, the trained model not only detects steel cracks but also predicts crack propagation, and supports rapid, on-site repair decisions.

#### 2.2.3. Integrated Frameworks

Data extracted from heuristic techniques or deep learning methods can be integrated into comprehensive frameworks for automated bridge inspection and the development of BrIM models. Perry et al. [66] proposed a streamlined bridge inspection system comprising advanced data analytics tools to automatically (1) generate a 3D point cloud model and segment structural elements using human-in-the-loop machine learning; (2) identify type, extent, growth, and 3D location of damages using computer vision techniques; and (3) establish a georeferenced, element-wise as-built BrIM model to document and visualize damage information (Figure 11).

Yamane et al. [67] developed an automated method to detect bridge damage and map it into a BrIM model. The technique combines DL for damage identification with SfM to estimate 3D coordinates of the damage. A YOLOv5 model was trained to recognize exposed steel rebars. It achieved an mAP50 of 64% for detection. The field test confirmed that the 3D coordinates and position data of the damage could be accurately reflected in the BrIM model. A comparison between the manually recognized damage and automatically detected 3D coordinates showed an error range between 4 and 15 mm. Additionally, by documenting the type and extent of damage, the workflow supports planning and prioritizing further repair interventions. The results indicated improved efficiency in a maintenance process based on inspection results (Figure 12).

### 2.3. Measurement of Structural Performance Parameters

Due to continuous exposure to severe loadings and extreme environmental conditions, the evaluation of key structural performance parameters, such as displacements, strains, and modal parameters of bridges is crucial.

#### 2.3.1. Displacements

Non-contact vision-based displacement measurement began in the 1990s with Stephen et al. [68], at the Humber Bridge in the UK. The most commonly used methods include Template-Matching algorithms [69,70,71,72], Optical Flow methods [73,74], and Digital Image Correlation (DIC) [75,76,77,78].

Template Matching involves locating a small portion of an image (known as the template) within the full image by sliding it across the scene and calculating similarity at each position. While accurate, this method is computationally intensive. To address this limitation, Wang et al. [72] introduced the Efficient Match Slimmer (EMS) algorithm combined with a robust Template-Matching system, Raspberry Vision. This cost-effective real-time bridge displacement measurement solution, integrated into the structure, was tested on two suspension bridges with object-to-camera distances of 133 m and 454 m, achieving maximum errors of approximately 1.4 mm.

Optical Flow methods are essential in computer vision for analyzing motion between sequential images. These methods calculate the apparent movement of brightness patterns and are typically represented as a vector field, where each vector shows pixel displacement between consecutive frames [79]. These methods assume brightness constancy and small inter-frame motion, allowing the use of differential techniques. There are several Optical Flow variants, including the Horn–Schunck, Farnebäck, and Lucas–Kanade methods. The last one is widely used because it is the most computationally efficient and is especially suitable for small-scale displacements, such as bridge monitoring. Based on this assumption, Hu et al. [80] evaluated pedestrian bridge vibrations to assess comfort criteria by integrating YOLOv5 for detecting pedestrian presence with an improved Lucas–Kanade method for the multitarget displacement tracking. A comparison between sensor data showed an error of only 2.3%, indicating the high precision of this approach.

Recent studies have explored the applicability of vision-based displacement monitoring in a real-world bridge environment. Bai et al. [81] conducted a comparative study evaluating different camera positions and platforms, including structure-mounted, UAV-based, and remote fixed position systems, for tracking bridge displacements under service loads. The methodology combined high-resolution imaging with Mask R-CNN for object detection and SIFT (Scale Invariant Feature Transformation) for feature matching and tracking. Dong et al. [82] employed consumer-grade cameras and computer vision algorithms to monitor the dynamic displacements of three suspension bridges. The study demonstrated the feasibility of long-distance, non-contact monitoring, with camera-to-target distance ranging from 1175 to 1350 m. The measured displacements were validated against conventional sensor data and FEM model predictions, showing strong consistency. These results show that vision-based approaches can serve as accurate, cost-effective alternatives for structural response monitoring.

The study conducted by Pan et al. [75] employed DIC, a technique that analyses variations in texture patterns or known markings on the object, to determine relative displacements. DIC is known for its simple experimental setup, minimal environmental requirements, and adaptability. Despite the attractive theoretical simplicity of DIC, displacement measurement processing is still computationally intensive. Over the past two decades, the algorithm has had significant improvements in both speed and accuracy. Its flexibility has made it applicable to different types of bridges [83], including concrete bridges [84,85], suspension composite steel–concrete solutions [76,86], masonry arch bridges [77,87], and steel bridges [78,88]. Koltsida et al. [89] were among the first to apply 2D DIC in the field to monitor a four-span masonry arch railway bridge under live traffic conditions. A single camera placed 10 m from the observed section recorded displacements during the passage of a 45 ton bogie (railroad truck) train. The horizontal and vertical displacements were measured in pixels using MatchID-2D DIC software. DIC proved to be suitable for crack identification, displacement measurements, and predicting structural response under different loads, as illustrated in Figure 13.

For long-span bridges, UAVs have become increasingly popular for capturing displacements in hard-to-access areas in a non-invasive and safe procedure. However, UAV ego-motions may result in reduced accuracy of displacement measurement [90] by interfering with structural motion. To mitigate this, several strategies have been proposed, such as (i) using digital high-pass filtering, (ii) using a stationary background object, and (iii) the integration of IMU [86,91,92]. In this context, using a UAV equipped with a commercial-grade video camera, Yoon et al. [93] simulated train passages in a laboratory experiment to measure the displacements of a railway bridge. By estimating the camera motion through background ROI and without relying on physical targets, they determined the 2D absolute displacement of the bridge. The Root Mean Squared Error (RMSE) was less than 2.14 mm, equivalent to 1.2 pixels in resolution.

While 2D vision methods can estimate dense motion fields, challenges arise when dealing with structures that contain multiple structural components with different lengths, surface orientations, and motion characteristics. To overcome this, researchers developed a vision-based dense 3D displacement measurement approach. This increased complexity but improved accuracy. Narazaki et al. [7,8] developed a vision-based dense 3D displacement measurement approach (Figure 14) capable of performing inference of displacements across all visible FEM nodes using a single camera.

The framework includes three steps: (i) generation of a synthetic environment, (ii) post-processing, and (iii) performance evaluation. A photo-realistic model simulates the full visual measurement process. Estimated displacements, obtained by applying post-processing algorithms to synthetic data, were compared with the ground truth displacement available from the FE analysis. The optimized algorithm involves (i) camera parameter estimation, (ii) camera motion estimation and compensation, (iii) 2D vision-tracking, and (iv) projection into 3D space. The study showed that a model-informed approach successfully estimated displacement across all visible nodes on a physically meaningful scale.

Despite their versatility, vision-based displacement measurement methods are sensitive to several external factors. Their accuracy may be affected by lighting conditions, camera vibrations, occlusions, and the need for a clear line of sight to the target points, especially in outdoor and large-span applications. These practical challenges must be considered when selecting the appropriate method for in situ bridge monitoring.

#### 2.3.2. Modal Parameters

Modal parameters, such as mode shapes, natural frequencies, and damping ratios, are important physical characteristics that reflect the structural condition [90]. For mode shape identification, several studies rely primarily on the Phase-Based Motion Magnification (PMM) algorithm [94,95]. PMM is an advanced video processing technique that amplifies small movements, making them visible to the naked eye [96]. The operation of PMM begins by decomposing the video into different spatial and temporal frequencies using techniques such as the Fourier transform, Empirical Mode Decomposition, and the Wavelet Transform. The phase of each frequency component is then analyzed. In this context, the phase refers to the position of a point within the wave cycle, which correlates with the motion in a video. Small phase changes are amplified to highlight movements. This amplification process is selectively applied to specific frequency ranges, while keeping other parts of the video unchanged. After amplification, the recorded video is reconstructed by combining the modified frequency components, resulting in visualized movements that were previously imperceptible. To evaluate the effectiveness of PMM, Chen et al. [9] tested the algorithm on cantilever beams to determine operational deflection shapes. A cantilever beam was instrumented with nine accelerometers, allowing a comparison between deflection shapes calculated from accelerometer data and those extracted from a high-speed camera. The results obtained were consistent, especially for higher frequencies, as shown in Figure 15.

A similar technique is Eulerian Video Motion Amplification (EVMA), which magnifies signal amplitude by analyzing pixel intensity changes over time. Eulerian Video Magnification (EVM) amplifies motion by processing sequences of video frames. It has been widely used to measure dynamic displacements and modal frequencies in structures. However, most applications remain limited to laboratory settings, comparing standard sensor data with EVMA outputs [9,97,98]. Shen et al. [99] extended this approach to both indoor and outdoor environments, successfully identifying mode frequencies and visualizing mode shapes of the structure.

Nguyen et al. [10] conducted a dynamic characterization of the Bo Nghi bridge in Vietnam and developed an FEM model that closely reflected reality. In their study, different damage scenarios were simulated, and their corresponding dynamic characteristics were calculated. These dynamic attribute-based damage indices were converted into images and used as input for a computer vision model to detect damage. To address challenges posed by modal identification of full-scale bridge structures, Hoskere et al. [100] proposed a method for extracting modal properties from UAV video footage. This technique corrects the UAV signal by (i) applying an adaptive scaling factor per frame, (ii) compensating for rolling shutter effects of CMOS sensors, and (iii) employing the NExT-ERA method, which cross-correlates relative displacements between the UAV-camera and the structure. The method was applied on a pedestrian bridge (Figure 16a) and successfully demonstrated that modal parameters extracted from UAV video matched those obtained from accelerometer data (Figure 16b).

#### 2.3.3. Strain Measurements and Stresses Estimation

Studies on stress monitoring in bridges often focus on estimating cable tensions and reinforcement bar strains, especially in cable-stayed bridges. Wangchuk et al. [101] proposed a UAV-based method to estimate cable tension by identifying the fundamental frequency under routine traffic loads. The process includes a 2 min video capture, RGB transformation and rotation, Template Matching, displacement calculation, FFT analysis, and final tension estimation. In laboratory tests using a vibrating table, the average frequency deviation between UAV measurements and the input frequency was 0.0022 Hz. Field tests on six cables with various lengths of the Rio Papaloapan Bridge showed a maximum error of 0.03 Hz (0.98%) when compared to accelerometer data. Notably, this method does not depend on camera calibration and supports near real-time monitoring.

Jana et al. [12] applied the Short-Time Fourier Transform (STFT) to determine the real-time frequency variation of a bridge cable on the Fred Hartman Bridge. Two locations were defined (locations 1 and 2), where the average error obtained was 17.05% and 28.72%, respectively. To reduce error estimation and avoid the installation of multiple sensors along the cable, the vibrational response was calculated using a phase-based motion estimation algorithm for each location. The resulting data were then combined to estimate the real-time frequency variation using a blind source separation (BSS) technique called Complexity Pursuit (CP). Finally, the cable tension was calculated in real-time based on the frequency history using the taut-string theory. The authors further proposed the BSS via the CP algorithm to merge vibrational responses from different locations, significantly reducing error. This demonstrates that combining multi-point data with advanced signal processing techniques can substantially improve the accuracy of real-time tension estimation.

Mirzazade et al. [102] developed a hybrid learning approach for non-contact prediction of strains in reinforced concrete (RC) girders, based on surface deformations monitored using Digital Image Correlation. The approach required a dedicated experimental dataset to train machine learning regression models. Laboratory results showed that the proposed method can reasonably predict strain on embedded reinforcements, offering new insights into innovative sensing applications with highly improved performance (Figure 17).

## 3. Big Data

Advances in remote bridge inspection and monitoring are supported by Big Data (BD). BD consists of collecting and storing a large volume and variety of data, which is processed at a high rate using technological tools and advanced analytical methods to predict patterns [103]. This section discusses the main features of BD, explored in Section 3.1, and their inclusion in bridge-related studies, presented in Section 3.2.

### 3.1. Main Features

Big Data is a collection of vast and diverse datasets for real-time processing and analysis. Within the context of bridge structures, BD is characterized by diversity, scale, and speed, which support both new bridge construction and the safe management of existing infrastructure. These features can be summarized by the 5Vs: Volume, Velocity, Variety, Veracity, and Value [13,103], as presented in Figure 18.

In bridge inspection, large **volumes** of data are produced annually due to the vast number of operational bridges. The use of UAVs and mobile inspection terminals has significantly improved inspection efficiency, resulting in a further increase in the amount of data generated. Each bridge SHM system generates a substantial amount of sensory data daily. Therefore, efficient management of this large amount of data, especially when integrating multiple SHM systems within the same region, remains a major challenge. Faced with these massive volumes of data, the rate at which data flows must also increase. Since data circulates continuously, its collection, processing, and analysis must be performed at the corresponding **velocity** to ensure the timely extraction of valuable information.

A wide **variety** of data types and formats fall under BD, including (i) unstructured data, such as inspection texts, images, and video footage; (ii) semi-structured data, such as tabular entries in inspection reports, and (iii) large-scale structured data, such as sensor outputs from SHM systems or data stored in relational databases. As these systems may be developed by different organizations, data formats are often heterogeneous, posing a significant challenge when attempting to integrate multiple systems. Therefore, the implementation of a unified data schema becomes essential but complex.

Big Data must not only be on a large scale but also reliable to provide actionable insights. **Veracity** involves both statistical reliability, that is, the consistency or degree of certainty of the data and the reliability of the data, which depends on origin, acquisition methods, processing, infrastructure, and computing facilities. Ensuring BD veracity guarantees authenticity, accuracy, and protection of data throughout its lifecycle, enabling correct interpretation despite increasing data variety, and source quantity.

The fifth characteristic, **value**, refers to the added value or utility that the collected data provides in decision-making, business operations, or analytical tasks. However, for data to offer value, it must be converted into knowledge. This is achieved through the combination of techniques such as data mining, predictive analytics, and text mining. This aims to achieve three major business objectives: cost reduction, quick and effective decision-making, and the development of new products or services.

### 3.2. Analysis Methods

Databases from diverse sources can be beneficial, providing complex and reliable information. However, understanding the interrelationships between multiple factors, such as the age of the bridge, structural type, design details, environmental conditions, lifetime loading, and deterioration patterns, can be difficult. This complexity surpasses the capacity of conventional statistical tools, suggesting the adoption of advanced, interdisciplinary, and computer-assisted processes, including data mining, machine learning (ML), and Knowledge Discovery in Databases (KDD). Among these, KDD has been the most widely applied in bridge-related studies [104,105,106], as it encompasses all phases of the data analysis process—from pre-processing to visualization of results—providing a more complete and coherent approach. The KDD process is defined as “the overall nontrivial process of discovering valid, novel, potentially useful, and ultimately understandable patterns in data” [104]. Data mining is a core component of this process and involves applying one or more ML algorithms to uncover patterns hidden within large data (Figure 19) [107,108,109].

The process begins with data selection, where relevant subsets are identified for analysis. Ensuring data quality and consistency leads to pre-processing, during which data is cleaned, inaccuracies are corrected, missing values are treated, and values are normalized to generate a suitable format for analysis. The data mining stage is central to the process, applying advanced techniques such as neural networks, Bayesian networks, or Decision Trees, to detect hidden patterns, relationships, or trends. Subsequently, these data undergo evaluation and interpretation, where patterns are analyzed in light of the study objectives and problem context. Finally, the knowledge management phase involves integrating these insights into decision support systems to ensure that knowledge gained during the process is integrated into information processing systems so that they are accessible and operationally useful.

To confirm the feasibility of using onboard systems for anomaly detection in railway bridges, a study analyzed track geometry data from several plate girder bridges on the Polish Railways (PKP PLK) [111]. Using the KDD framework, vertical irregularities in the rails were examined to interpret the structural response of the bridge and its variation over time. In another study, inventory and five-year inspection data of 2849 slab, girder, and box-girder bridges in south-central Taiwan were analyzed using the KDD to uncover deterioration patterns. Bridges were clustered and examined to identify associations between inventory attributes and component deterioration. This process generated association rules indicating which bridge configurations are more prone to degradation of a specific component [112]. The KDD procedure in these examples follows a structured path: defining a research objective, collecting references, inputting inventory data, performing initial processing and categorization, applying clustering techniques, integrating inspection data, and conducting a final analysis to interpret bridge performance.

## 4. Digital Twins

Digital Twins in bridge inspection create virtual models of physical bridges, fed by real-time data from sensors and other sources. These models facilitate the monitoring of structural integrity, early detection of potential issues, and simulation of various exogenous factors, such as weather and traffic. By continuously integrating real-world inputs, DTs enhance predictive maintenance and reduce the need for costly manual inspections. Additionally, DTs improve engineer’s safety by validating repair strategies in a virtual environment before implementing them on-site. Compared to traditional SHM methods that rely on isolated data streams, DTs provide an integrated framework that combines monitoring, predictive analyses, and visualization. This enables more informed decision-making and proactive maintenance strategies, although implementation remains mostly limited to high-value assets due to cost and system complexity. To clarify how this technology works, this section begins with an overview of the concept (Section 4.1), followed by a description of the typical architecture (Section 4.2), and recent applications (Section 4.3).

### 4.1. Definitions and Concepts of Digital Twin

There are several interpretations and definitions of DTs, reflecting their importance and relevance across multiple domains and industries. Despite these nuances, most definitions share a common set of theoretical principles and essential components. Table 2 compiles representative definitions of DTs found in the literature, highlighting the key elements that underpin each description.

DTs have emerged as an innovative technology in asset management and lifecycle processes, particularly during the design, construction, and maintenance phases [124]. While most existing studies emphasize the use of DTs during the operation and maintenance phase, their application during the construction phase remains relatively underexplored. The construction phase presents unique challenges to effective DT implementation due to its complexity, involvement of multiple stakeholders, and limited digitalization of processes [125]. Nonetheless, integrating DTs across all stages of asset development is beneficial. Hoskere et al. [126] provided a comprehensive review of DT technologies for bridge infrastructure, presenting a structured overview of DT concepts and a cohesive framework for DT design and implementation.

### 4.2. Digital Twin Architecture: Data Acquisition, Processing, and Visualisation Modules

The architecture of DTs comprises three main modules [127,128]: (i) data acquisition and control; (ii) advanced simulation, optimization, and predictive analytics; and (iii) a visualization interface. The data acquisition and control modules are integrated into the physical asset to manage data collection from various sensors. This module ensures the accuracy and consistency of acquired data, enabling real-time analysis and informed decision-making. Moreover, the integration of different sensor types allows comprehensive and multidimensional monitoring of environmental and structural conditions. UAVs equipped with computer vision systems expand the monitoring scope, supporting inspection of hard-to-reach areas and capturing high-resolution data. Meanwhile, static sensors provide continuous and long-term data, essential for trend analysis and assessment of structural integrity. The seamless integration of these systems through the data acquisition module is key to developing a robust DT platform, enabling a real-time response to dynamic variables.

Collected data is transmitted via IoT/Big Data technologies and protocols to the DT platform. This data, in continuous or periodic flow, is organized using appropriate ontologies and semantics, and is stored, primarily in the cloud, to handle large volumes. In the analytics module, knowledge extraction is performed using dedicated routines and algorithms, including (i) scan-to-BrIM, transforming point clouds into object-based models; (ii) AI classifiers for sensor data and image data, supporting degradation risk assessment and anomaly detection; and (iii) predictive analytics, forecasting asset performance based on historical trends. The visualization interface enables advanced graphical representation, which is tailored to user profiles and supports decision-making.

### 4.3. Applications of Digital Twins in Bridge Management

Digital Twins, in combination with BrIM, have been integrated into Bridge Management Systems (BMS) to create detailed digital models of bridges, including precise information on geometry, materials, and structural systems. The Internet of Things (IoT) and BD serve as critical enablers of this approach, supporting continuous data acquisition from the physical environment, as well as ML for predictive analysis and anomaly detection. The synergy between these subsystems significantly enhances the capabilities of DTs in operational decision-making, while promoting improved social and environmental sustainability. Given their benefits, DTs have received increasing attention in bridge research. Jasiński et al. [129] combined load testing, BrIM modelling, and FEM modelling to create and validate a DT for a simple bridge case. In this example, the BrIM model (Figure 20a) provided the foundational geometry of the structure, upon which the FEM model was constructed and embedded within the BIM environment. The FEM model allowed the complex bridge geometry to be discretized into smaller elements, each with specific material properties, loads, and boundary conditions. Both static and dynamic load tests were performed to calibrate and validate the DT (Figure 20b), ensuring an accurate representation of the bridge structure.

With the advancement of sensing and IoT technologies, the vast amount of heterogeneous data generated from regular inspections and real-time monitoring presents a complex challenge for the synchronization of DTs. In bridge applications, restricted communication can further complicate implementation. Most existing bridge DTs are cloud-based and depend on stable connectivity, often overlooking system resilience during temporary communication loss. In this context, Gao et al. [130] proposed an IoT-based communication framework to mitigate time delay issues. This structure was implemented in a prototype through cross-platform integration and validated across various scenarios. Results demonstrated superior performance, low latency, and excellent fault tolerance compared to other existing DT systems, contributing to enhanced bridge safety and operational efficiency (Figure 21).

For large bridge structures such as cable-stayed and suspension bridges, Shim et al. [131] proposed and developed a DT model using data collected from sensors. The process involved organizing relevant information for each structural member. The first model represented a stiffening girder in a suspension bridge, followed by a suspension bridge model. The methodology proved robust enough to cover two distinct bridge types. Although the influence of local damage on the global or individual member behavior was not fully established, the monitoring data allows for the model to update when new damage is detected.

Chacón et al. [132] presented developments in the DT of bridges in a high-speed train network in Spain. The study consisted of deploying multiple data-collection techniques, primarily from load tests and reality capture. Bridges were geometrically virtualized using IFC Standards, and numerical simulations were performed for all scenarios tested during static and dynamic load evaluations. All measurements were transmitted via IoT and visualized within a Common Data Environment (CDE). The study demonstrated how efforts in sensing, simulation, modelling, assessment, and validation can be systematically merged for ongoing use during the regular operation of the asset. Figure 22 presents how the DT platform acts as a data integration hub, consolidating inputs from measurements, BrIM, simulations, and more, while facilitating information exchange among stakeholders during the asset’s lifecycle.

Perry et al. [66] proposed a DT framework with multiple applications to enhance bridge assessment and management. The DT integrates tools to store, visualize, and analyze data from UAV-enabled remote inspection and computational models to support maintenance decision-making. Key components include a data analysis module and a model library. The analysis module employs AI and computer vision to identify the location, extent, and progression of defects, as well as detect structural components and connections from imagery. Additionally, three-component (3C) dynamic displacements are measured from UAV videos. The model library assesses structural performance using (1) a visualization model for location-based data queries, (2) an automatically generated FEM model for simulation, and (3) a surrogate model for rapid prediction of behavior. The DT evolves as new data becomes available, updating its models and suggesting executable actions for improved monitoring and repair.

## 5. Augmented Reality

Infrastructure inspectors assess structural changes across time and space using their experience and technical resources, such as drawings, images, and inspection records, while contextualizing their findings to support decision-making processes [28,35]. The American Society of Civil Engineers (ASCE) predicts that “*In 2025, intelligent infrastructure (embedded sensors and real-time onboard diagnostics) will have transformed the landscape through the rapid advancement and adaptation of high-value technologies throughout a structure’s lifecycle.*” Real-time monitoring, sensing, data acquisition, storage, and modeling have significantly enhanced predictive capabilities and will lead to more informed decisions. Furthermore, the ASCE highlights that engineers will “*rely on and leverage real-time access to dynamic databases, sensors, diagnostic tools, and other advanced technologies to ensure informed decisions are made*”. Digitalizing the interface between inspectors, collected data, and end-users can be transformed through the application of Augmented Reality (AR). Recently, AR has been increasingly employed in infrastructure-related applications, improving accuracy, usability, and efficiency [133].

The following sections describe the key components of AR systems and provide examples of their implementation in the SHM of infrastructures. Section 5.1 presents a brief description of the state of the art, followed by an overview of AR capabilities in Section 5.2. Section 5.3 discusses the adoption of AR technologies through recent case studies.

### 5.1. Augmented Reality Framework for Bridge Inspection

AR refers to the enrichment of the real world with virtual elements. In Virtual Reality (VR), the real world is replaced by simulated environments. AR is more widely applied in practical cases than VR, as it allows for real-time interfacing between digital drawings and physical structures during inspections. Inspectors can access real-time, relevant digital information to assist in the inspection [134]. Within today’s inspection workforce, complexity exists as follows: (1) Trained inspectors can prioritize which data to collect to assess structural change and damage. However, the human factors and processes still rely on previous experiences, labor hours, and traditional forensic structural engineering knowledge; (2) data collection remains limited to the human perception and must be manually logged, often constrained by accessibility and the inspector’s experience, resulting in time-intensive and potentially inconsistent outputs; and (3) consequently, inspection-related decisions are slow, qualitative, and largely dependent on human judgement.

### 5.2. Capabilities, Limitations, and Technological Advances in Augmented Reality

AR initially enabled users to interface with 3D drawings in real-time through a basic integration of hardware and software. Early challenges included a limited field of view and visual resolution [135], but later developments addressed these, improving the realism of holographic and physical object interaction [135,136]. In the early 2000s, hardware innovations such as Google Glass, Moverio smart glasses, and Microsoft HoloLens revived AR adoption. The HoloLens features a Holographic Processing Unit (HPU) that builds a 3D model of the environment assisted by spatial cameras, IMU, depth cameras, and other sensors installed in the system [137,138].

AR enhances learning through interactivity, feedback, and immersion. Reviews show a positive impact of AR on learning outcomes, including motivation, satisfaction, engagement, and attitude. However, limitations remain, including usability issues, cognitive load, and instructional design challenges [139,140,141]. Catbas et al. [142] highlighted how the next generation of AR/VR may allow remote bridge inspections and health monitoring, eliminating physical presence requirements and lowering health risks and costs.

### 5.3. Emerging Applications of Augmented Reality

On March 1st, 2019, a workshop was held to identify areas where AR research could support challenges faced by bridge inspectors. Department of Transportation (DOT) attendees identified an application enabling bridge inspectors to automatically measure crack widths in the field and compare them across periodic inspections [143,144,145,146]. Ongoing efforts from the railway sector, the air force, NASA, NCHRP, and emergency services highlight the practical potential of AR for structural damage detection and assessment [147,148]. Recent studies have applied an AR camera (ARCam) to inspect the column anchor bolt positions before steel column installation and to assess plumbness after assembly [149]. The ARCam demonstrated time-related advantages when compared to conventional total stations, thus increasing productivity and reducing cost, while maintaining measurement precision within standard tolerances.

Another application of AR for structural damage assessment involves integrating wireless sensors with AR systems. This connection enables sensor data to be processed using algorithms and computing tools, allowing SHM information to be overlaid into the inspector’s field of view via an AR device. In the study by Fawad et al. [150], three wireless sensors (WSG, MA, and WMS) representing a full SHM system transmitted data to an IoT web platform, which was then linked to an AR application developed in Unity 3D. This enabled real-time visualization, analysis, storage, and sharing of SHM data through the HoloLens (Figure 23), confirming that internal forces, displacements, and stresses remained within Eurocode limits.

Recent studies have also explored the ability of AR to visualize structural vibration modes [15,16]. Carter et al. [16] created a framework displaying real-time time-domain and frequency-domain records along with modal shape estimations, via Wi-Fi using the Python library CESSIPy. The system’s performance was negatively impacted as the number of plotted points or sampling frequency increased. Cuong et al. [151] presented a BIM-based application in which HoloLens is used to support the management of bridge inspections and maintenance from the office. The application includes modules that enable users to examine and update the ongoing inspection and maintenance activities.

## 6. Conclusions and Recommendations

This paper has reviewed the state of the art of emerging methodologies for visual inspection, particularly those based on computer vision and supported by ML algorithms and Big Data analytics. These approaches enhance the efficiency and reliability of inspections by acquiring data beyond the capabilities of traditional methods and providing real-time feedback through Augmented Reality features.

This review highlights recent developments in remote and autonomous bridge inspection technologies, supported by case studies that validate their effectiveness. It begins with computer vision techniques, addressing 3D reconstruction, damage detection, and physical measurements. However, such methods generate large datasets that must be organized and structured. Thus, this paper introduces the concept of Big Data, including the key components required for effective data handling.

With the growth of digital construction and data analytics, the development of DTs has emerged as a significant advancement in civil engineering. DT offers real-time mapping of physical infrastructures, providing insights for maintenance, decision-making, and control. This review outlines DT architecture and its application in bridge engineering, while also linking its evolution to advancements in AR and VR, particularly in inspection tasks.

While many of these technologies show great promise, they differ significantly in terms of technological maturity, ease of integration into existing Bridge Management Systems, and sensitivity to field conditions. Methods such as photogrammetry and heuristic detection are already being implemented in practice, whereas advanced approaches like Digital Twins or NeRF-/GS-based reconstruction are still largely experimental and often require substantial infrastructure and expertise. These differences are important when evaluating the suitability of each method for real-world applications in bridge inspection and maintenance.

Future developments in infrastructure inspection are expected to be shaped by the transition from Industry 4.0 to Industry 5.0, which emphasizes human–machine collaboration and the integration of intelligent systems. In the context of bridge inspection, systems like Motion Capture (MoCap) systems may enhance and optimize the movements of workers, machinery, and inspectors on-site. By collecting real-time data on task execution, these devices support improved safety management, ergonomic assessment, and operational efficiency. Moreover, MoCap can support the creation of realistic simulations for training purposes and the rehearsal of complex tasks in virtual environments, reducing on-site risks.

Despite their benefits, these emerging procedures require integration into existing bridge inspection frameworks. It is essential to ensure that computer vision-based inspections are at least as reliable as those conducted by human inspectors. Furthermore, there is the question of whether inspection reports and databases should be adapted to integrate new elements obtained with these new methods. These are issues that are currently studied by road/railway authorities, especially as they are not their usual area of expertise.

## Figures and Tables

**Figure 1 sensors-25-05708-f001:**
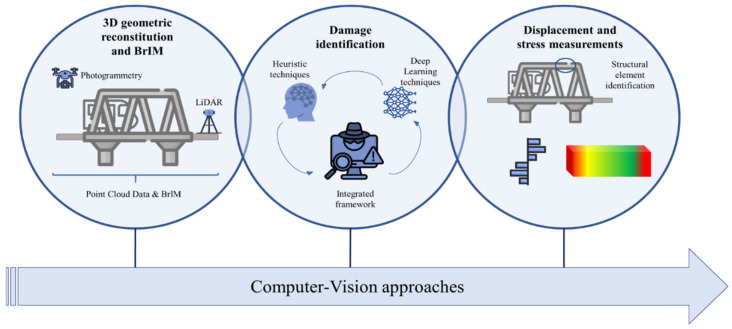
Computer vision approaches.

**Figure 2 sensors-25-05708-f002:**
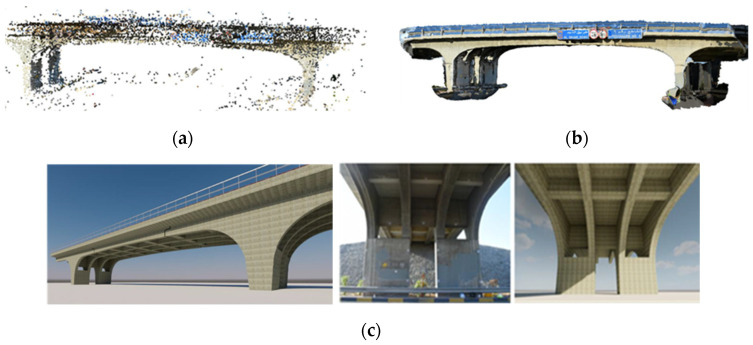
Generated 3D model of the bridge: (**a**) merge of four-point cloud data, (**b**) photo modeler 3D model, (**c**) Revit 3D model (adapted from [2]).

**Figure 3 sensors-25-05708-f003:**
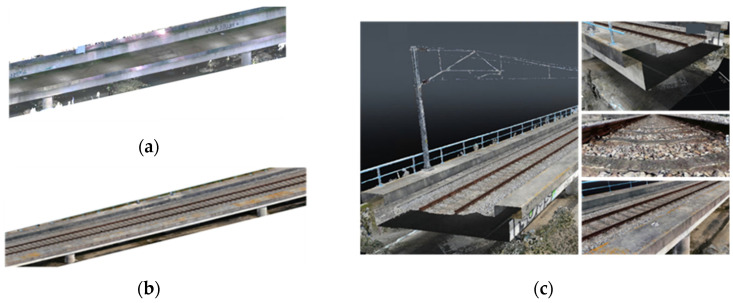
Reality capture of access viaduct of Pirâmides railway bridge: (**a**) TLS model, (**b**) SfM-based model based on UAV data, and (**c**) hybrid model (adapted from [28]).

**Figure 4 sensors-25-05708-f004:**
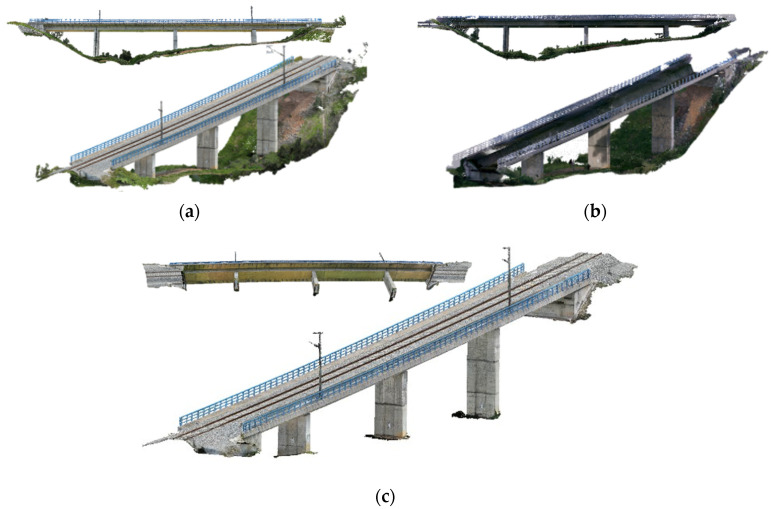
3D point cloud model: (**a**) photogrammetry, (**b**) LiDAR, and (**c**) fusion strategy [29].

**Figure 5 sensors-25-05708-f005:**
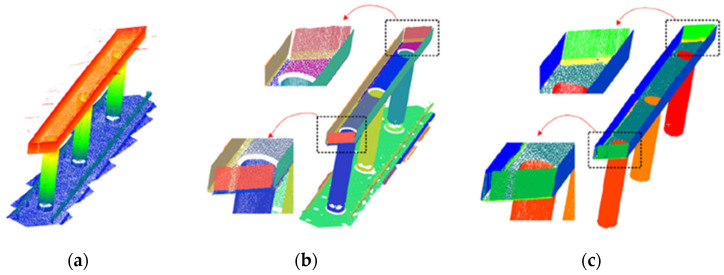
Comparison between surface extraction results: (**a**) sub-dataset of a pier, (**b**) surface extracted by RANSAC, and (**c**) surface extracted by the method proposed by Hong et al. [25].

**Figure 6 sensors-25-05708-f006:**
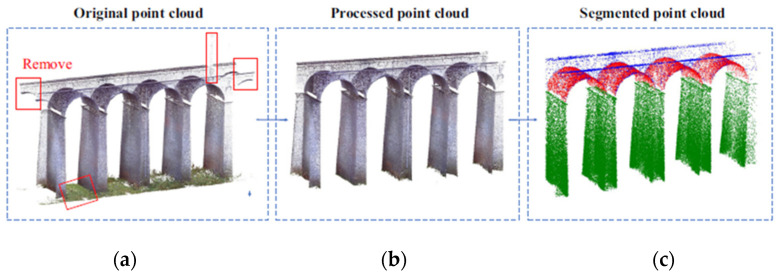
Point cloud processing: (**a**) original point cloud, (**b**) background removal, and (**c**) semantic segmentation [36].

**Figure 7 sensors-25-05708-f007:**
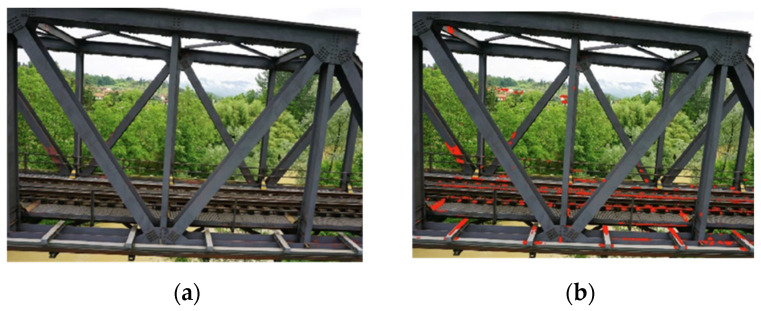
Application of heuristic filters to the case of a steel railway bridge: (**a**) original image, (**b**) defect detection (adapted from [49]).

**Figure 8 sensors-25-05708-f008:**
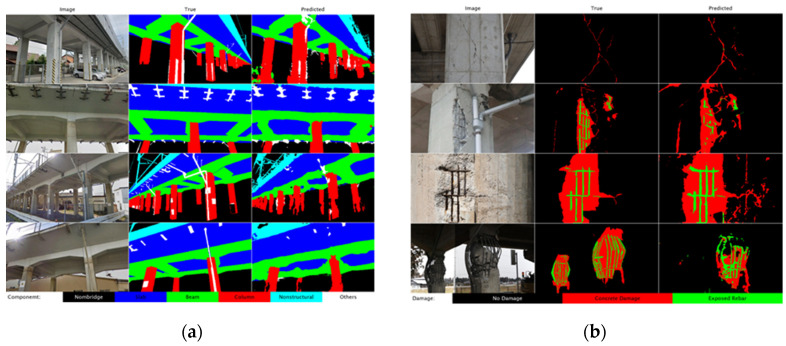
Example of visual recognition results presented by Narazaki et al. [21]: (**a**) structural components and (**b**) structural damage.

**Figure 9 sensors-25-05708-f009:**
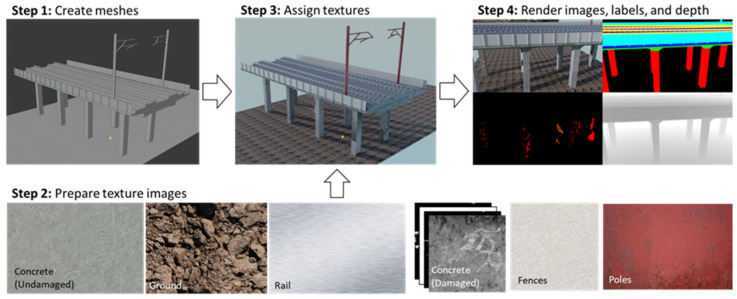
Steps to create photo-realistic synthetic environments [21].

**Figure 10 sensors-25-05708-f010:**
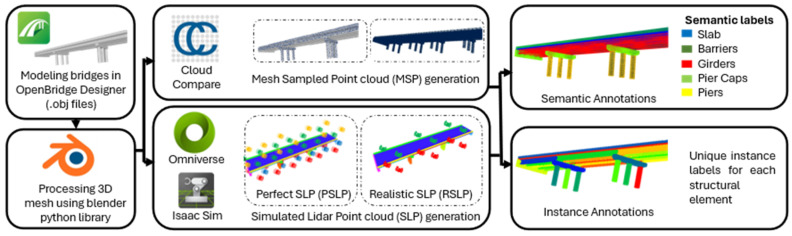
Virtual LiDAR-based synthetic bridge point clouds generation processes and automated annotation for DL-based 3D instance segmentation of highway bridges [64].

**Figure 11 sensors-25-05708-f011:**
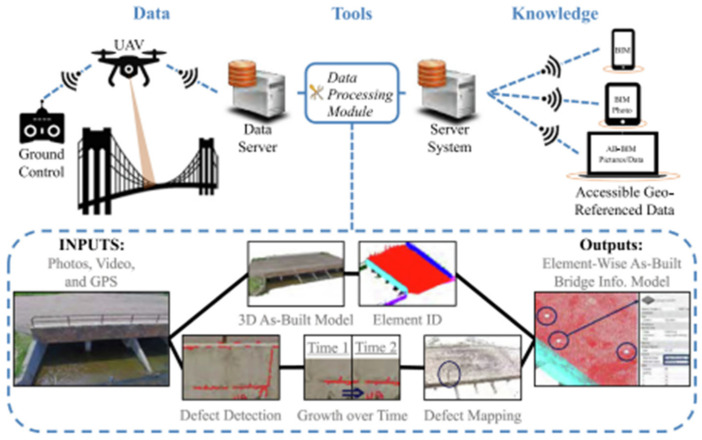
Streamlined bridge inspection system [60].

**Figure 12 sensors-25-05708-f012:**
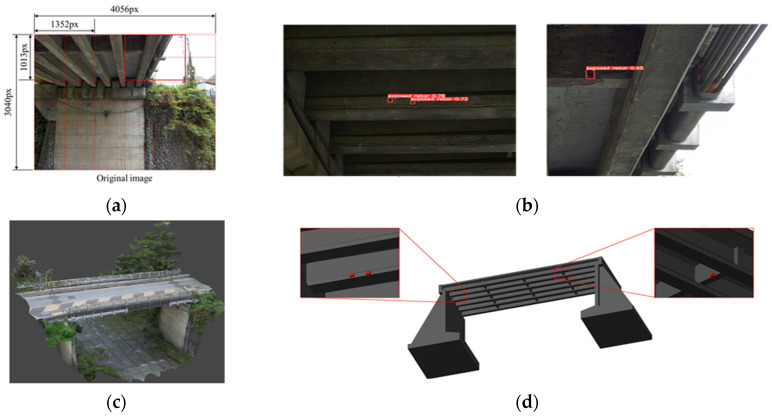
Integrating damage into BrIM: (**a**) damage area with crop representation (red lines), (**b**) full 3D model using SfM, (**c**) detection results with cropped images, and (**d**) BrIM model with damage labels [67].

**Figure 13 sensors-25-05708-f013:**
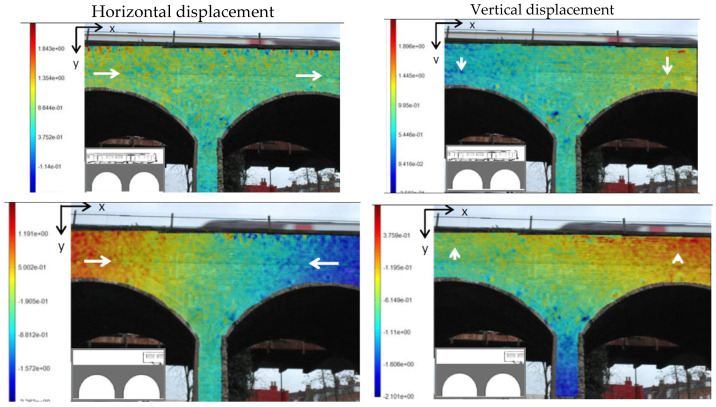
Horizontal and vertical displacements measurementusing a DIC-based method (adapted from [89]). White arrows indicate the displacement direction.

**Figure 14 sensors-25-05708-f014:**
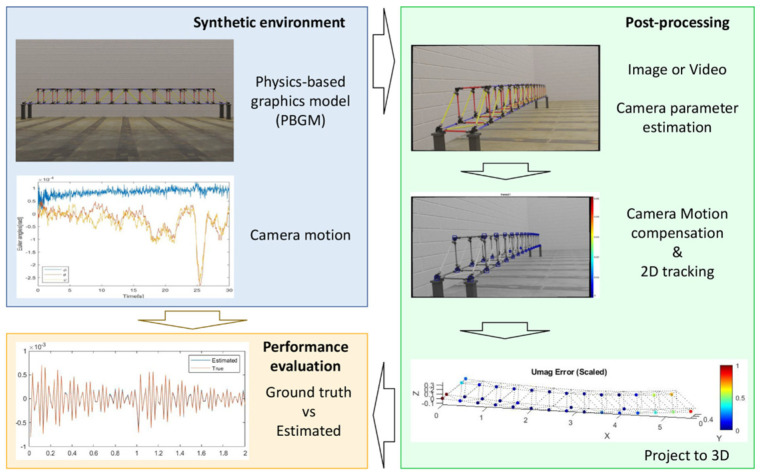
Computer vision-based dense 3D structural displacement measurement framework [8].

**Figure 15 sensors-25-05708-f015:**
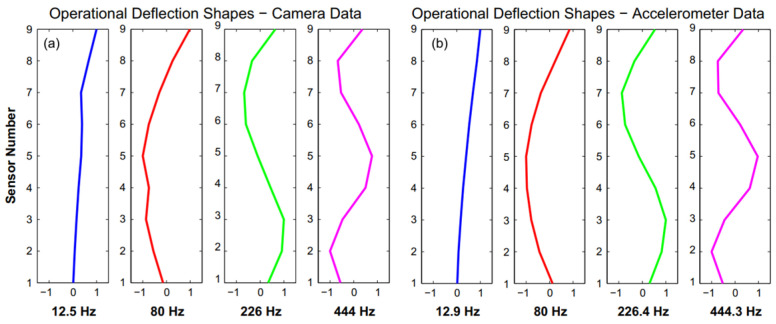
Normalized operational deflection shapes of the cantilever beam from (**a**) displacements extracted from camera video and (**b**) accelerometer data [9] (with permission from ASCE).

**Figure 16 sensors-25-05708-f016:**
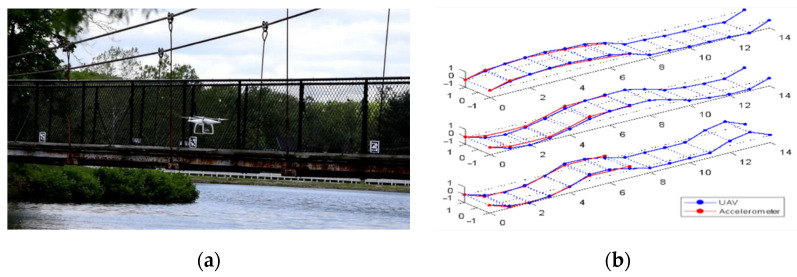
Remote modal identification of a suspended pedestrian bridge: (**a**) image of UAV Phantom 4 recording video and (**b**) mode shapes [100].

**Figure 17 sensors-25-05708-f017:**
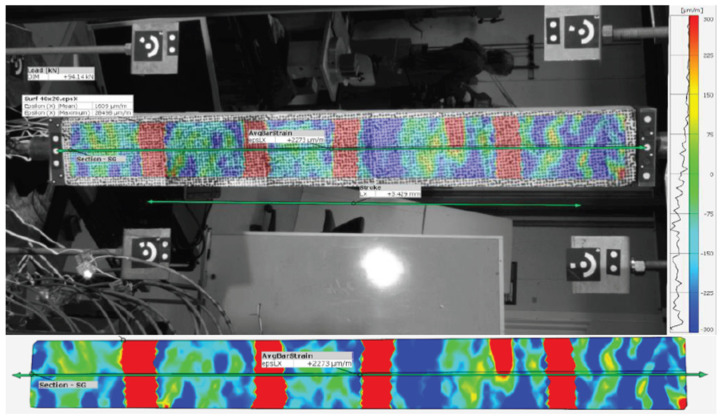
Monitored surface strain and displacement on a reinforced concrete sample [102].

**Figure 18 sensors-25-05708-f018:**
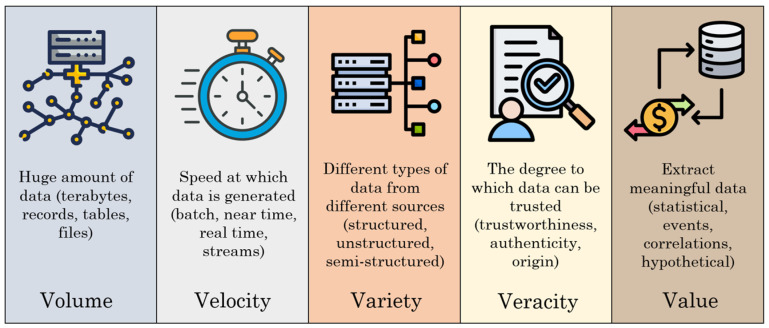
Characteristics of the 5Vs of Big Data.

**Figure 19 sensors-25-05708-f019:**
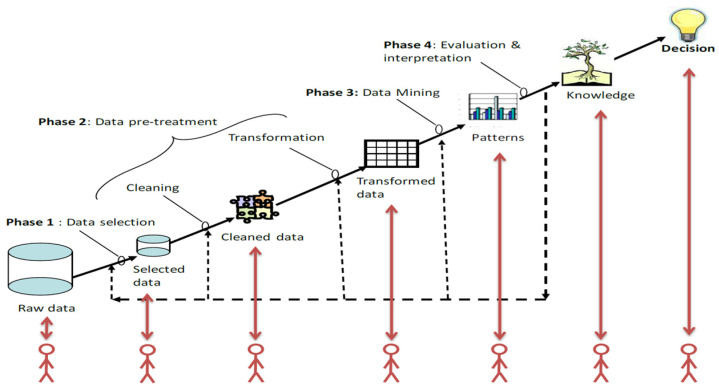
Knowledge Discovery in Databases (KDD) process [110].

**Figure 20 sensors-25-05708-f020:**
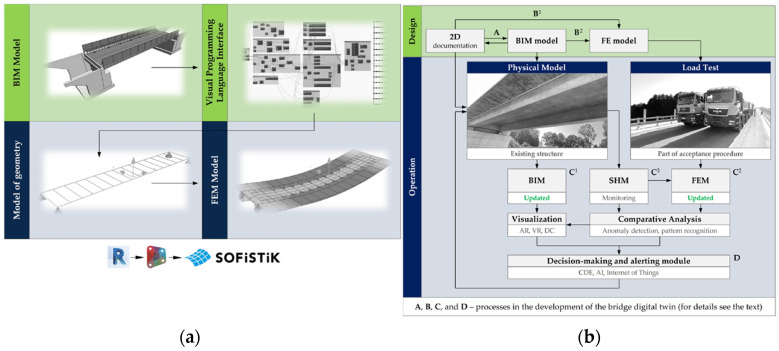
(**a**) BrIM model, and (**b**) Digital Twin of the bridge with BrIM model, FEM model, and load test [129].

**Figure 21 sensors-25-05708-f021:**
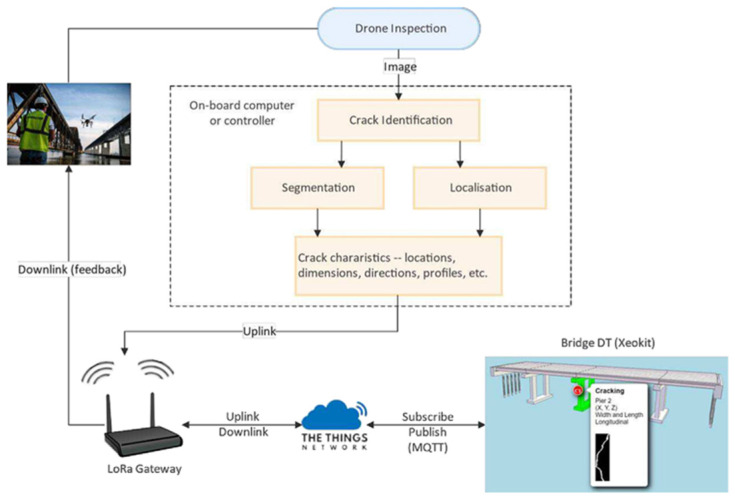
Developed bridge DT prototype for drone-enabled bridge inspection [130].

**Figure 22 sensors-25-05708-f022:**
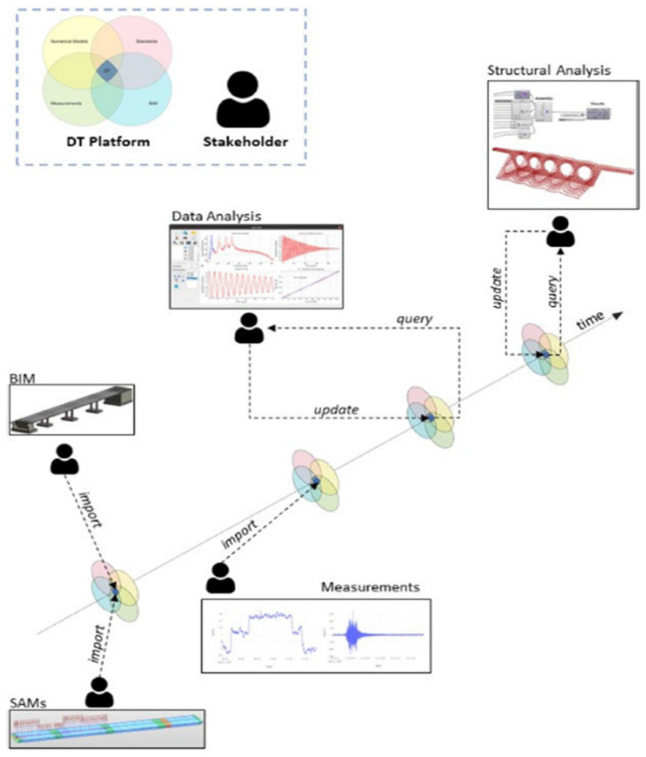
DT platform: timeline of the interactions regarding measurements, simulation, standards, and BIM [132].

**Figure 23 sensors-25-05708-f023:**
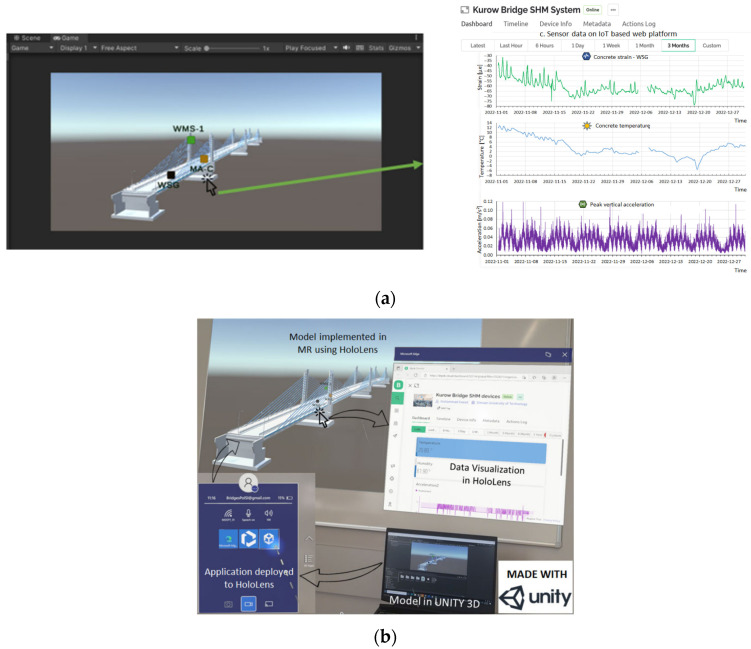
(**a**) AR application: app development in UNITY 3D and its linkage with the IoT platform, and (**b**) visualization of SHM data in HoloLens [150].

**Table 1 sensors-25-05708-t001:** Methodologies used for the Remote Inspection of Bridges.

Methodologies	Application	Techniques
Computer Vision	3D geometric reconstitution	Photogrammetry (SfM—Structure from Motion, MVS—Multi-View Stereo) LiDAR (ToF—Time-of-Flight) Hybrid strategies (ICP—Iterative Closest Point) AI-Based (NeRF—Neural Radiance Fields, RANSAC, PFM—Plane Fitting Method, Gaussian Splatting, CNN, and Autoencoder)
	Damage identification and component detection	Heuristic (edge detection filters) Deep learning (CNN, Mask-R-CNN, YOLO) Integrated frameworks
	Measurement of strains, displacements, and modal parameters	Template-Matching algorithms Optical Flow methods Digital Image Correlation (DIC) Phase-Based Motion Magnification (PMM) Eulerian Video Motion Amplification (EVMA) NExT-ERA UAV motion subtraction: digital filters, stationary background target, IMU
Big Data	Robust knowledge of the asset’s structural behavior from varied data	Knowledge Discovery in Databases (KDD) Neural networks Bayesian networks Decision Trees
Digital Twins	Virtual representation of real behavior of a physical asset	IoT Data structuring techniques Common Data Environment (CDE) Scan-to-BrIM BrIM AI Optimization techniques Simulation (FEM, DEM, others) Predictive analysis
Augmented Reality	Interface between drawings and real structures during inspections	

**Table 2 sensors-25-05708-t002:** Definitions of DT.

Reference	Definition of DT
[113]	Comprehensive physical and functional description of a component, product, or system, including all potentially useful information across current and subsequent lifecycle phases
[114]	A digital informational construct about a physical system, created as a standalone entity and linked to the physical system throughout its lifecycle
[115]	A digital model that dynamically represents and mimics an asset’s real-world behavior; built on data
[116]	Cyber representation within Cyber-Physical Systems, composed of multiple models and data
[117]	Enabled Big Data analytics, faster algorithms, increased computation power, and data availability, allowing for data-enabled real-time control and optimization of products and processes
[118]	An integrated, multi-physics, multiscale, and probabilistic simulation of a complex product that uses the best available models, sensor updates, etc., to mirror the life of its corresponding twin
[119]	Using a digital copy of the physical system to perform real-time optimization
[120]	Composition of disparate digital models which gives rise to a higher fidelity model of a product
[121]	Continuous interactive process between the physical manufacturing facility and its digital counterpart
[122]	Virtual representations of physical entities
[123]	A model where each product is also directly connected with a virtual counterpart

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
