# Peer review of "Methodologies for Remote Bridge Inspection—Review"

_sensors, 2025, doi:10.3390/s25185708_

Round 1

Reviewer 1 Report

Comments and Suggestions for Authors

Minor revision
This review covers a wide range of remote bridge inspection methodologies, with a strong focus on computer vision, big data, digital twins, and AR. The topic is timely and the structure is logical, but some key details and critical analysis are missing. The manuscript reads more like a technical catalog than a critical review, and some sections lack depth or clarity.
(1)The review mostly summarizes methods—can you add more critical comparison between them?
(2)Can you clarify the novelty of this review compared to other recent reviews on remote bridge inspection?
(3)How were the papers selected for inclusion? Is there a systematic review process or just a narrative summary?
(4)The “Error! Reference source not found.” appears many times.
(5)Are there any limitations or challenges for each method? For example, what are the main drawbacks of vision-based displacement measurement?
(6)The section on AI-based 3D reconstruction mentions NeRF but says no bridge applications exist; why include it, and are there related methods with actual bridge use?
(7)Some case studies are described in detail, others are just mentioned—can you balance this or explain why?
(8)The review mentions “Industry 5.0” and MoCap in the conclusion, but these are not discussed in the main text. Should this be expanded or removed?
(9)How do these methods compare in terms of cost, required expertise, and scalability for real-world bridge management?
(10)The writing is sometimes repetitive, especially in the introduction and summary.
(11)For CV inspection, please refer to 3D vision technologies for a self-developed structural external crack damage recognition robot; Automation in Construction.
(12)For Lidar & Remote sensing, please refer to Dual-Frequency Lidar for Compressed Sensing 3D Imaging Based on All-Phase Fast Fourier Transform. J. Opt. Photonics Res. 

Author Response

Thank you to the reviewer for the pertinent comments.
All questions have been answered with great care and can be verified in the attached Word document. I would also like to mention that the English was reviewed again by four different co-authors to ensure that the final result was the best possible.

Reviewer 2 Report

Comments and Suggestions for Authors

A good and comprehensive review. The link to the references must be corrected.

Author Response

Thank you to the reviewer for the pertinent comments.

All questions were answered with great care and all references were corrected as requested.

I would also like to mention that the English was reviewed again by four different co-authors to ensure that the final result was the best possible.

Reviewer 3 Report

Comments and Suggestions for Authors

This paper is a survey of using remote sensing technologies and methodologies to inspect bridges. It is a time-consuming job, but the authors did their best. The following concerns should be addressed to improve this research:

1) There should be a section particularly designed for the setup and deployment of these devices and their performance limitations should be covered for the remote bridge inspection, based on the review. This knowledge is quite useful for people who are interested in this topic.

2) Please provide the knowledge to distinguish the remote bridge monitoring and the remote inspection.  Provide the connection to the bridge inspection using remote bridge inspection.

3) There are too many abbreviations lacking explanation before their use in Table 1. The same happened to GCP, UAV, CAD, ReLU, GPU, TPU, BIM, RMSE, etc.

 4) The text font and sizes in figures should be consistent and large enough to be readable (Figures 1, 9, 10, 11, 14, 17, 18, 19, 20, 21, 22). Figures should be explained in the text paragraphs, especially those using different colors in the figures (Figures 2, 5 ,6,8, 9).

5) Why are the first letters of big data and digital twins capitalized in the Abstract?

6) The No.15 author's address is incorrect.

7) The title of Section 2 should be vision-based technologies or methodologies.

8) Please address these questions: (a) Why were vision-based technologies used in bridge inspection? Their purposes are for the integrity or serviceability of bridges.

9) Figure 1 is oversimplified.

10) Check the section indexes (e.g.,2.1., 2.1.1.) are the required format of this journal.

11) If possible, authors should include more research and applications of remote bridge inspection. For example, regarding to vision-based technologies, our research related to using various camera placements to monitor and assess pedestrian, railway, and traffic bridges could be good examples: (a) https://doi.org/10.1007/s13349-023-00720-6; (b) https://www.mdpi.com/1424-8220/23/19/8161.

12) Citation problem: (a) Citations for Revit and ReLU are missing. (b) Citation for the Lines from 206 to 210 is missing for NeRF. I am wondering why the applications of Gaussian Splatting are not reviewed.

13) In Sections 4 and 5, the titles for each subsection are too short to direct readers well to understand the authors' point.

14) Line 163, the author's name is wrong and different from the same citation in the following section. Please check all the citations accordingly.

Author Response

(The authors gave the same response as above.)

Round 2

Reviewer 3 Report

Comments and Suggestions for Authors

Thanks for the authors' contribution to this field.